# Unravelling the structural complexity of glycolipids with cryogenic infrared spectroscopy

Carla Kirschbaum [1,2], Kim Greis[1,2], Eike Mucha[2], Lisa Kain[3], Shenglou Deng[4], Andreas Zappe[1], Sandy Gewinner[2], Wieland Schöllkopf [2], Gert von Helden [2], Gerard Meijer [2], Paul B. Savage[4], Mateusz Marianski [5,6], Luc Teyton[3] & Kevin Pagel [1,2✉]

Glycolipids are complex glycoconjugates composed of a glycan headgroup and a lipid moiety. Their modular biosynthesis creates a vast amount of diverse and often isomeric structures, which fulfill highly specific biological functions. To date, no gold-standard analytical technique can provide a comprehensive structural elucidation of complex glycolipids, and insufficient tools for isomer distinction can lead to wrong assignments. Herein we use cryogenic gas-phase infrared spectroscopy to systematically investigate different kinds of isomerism in immunologically relevant glycolipids. We show that all structural features, including isomeric glycan headgroups, anomeric configurations and different lipid moieties, can be unambiguously resolved by diagnostic spectroscopic fingerprints in a narrow spectral range. The results allow for the characterization of isomeric glycolipid mixtures and biological applications.

---

[1] Institut für Chemie und Biochemie, Freie Universität Berlin, Berlin, Germany. [2] Fritz-Haber-Institut der Max-Planck-Gesellschaft, Berlin, Germany. [3] Department of Immunology and Microbiology, Scripps Research, La Jolla, CA, USA. [4] Department of Chemistry and Biochemistry, Brigham Young University, Provo, UT, USA. [5] Department of Chemistry and Biochemistry, Hunter College, The City University of New York, New York, NY, USA. [6] The PhD Program in Chemistry, Graduate Center, The City University of New York, New York, NY, USA. ✉email: kevin.pagel@fu-berlin.de

Glycolipids are amphiphilic biomolecules that are omnipresent in the cell membranes of all kinds of organisms ranging from bacteria to humans[1]. Playing key roles in cellular interactions and signal transduction, they are essential for the development and function of multicellular organisms[2–4]. Furthermore, immune responses can be modulated by α-linked glycolipid antigens such as α-galactosyl ceramides (GalCer). In case of microbial infections, they trigger the activation of natural killer T-cells (NKT), a T cell subset sitting at the interface between innate and adaptive immunities[5–8]. α-GalCer was first isolated as an antitumor agent from marine sponge[9] and was thought to be produced exclusively by bacteria and porifera[10]. In mammalian cells, however, only β-isomers were detected—which raised the question how NKT cells are triggered in mammals[5,11]. At the end of a controversial debate lasting for more than a decade, the presence of low-abundant, endogenous α-GalCer in mammalian cells was finally revealed using a combination of biological, enzymatic, and immunological assays[5,12] and was later confirmed by direct biochemical evidence[13]. On the other hand, established analytical techniques failed to detect α-GalCer in the presence of highly abundant, completely non-antigenic β-GalCer, which caused severe confusion in the meantime[14].

The cumbersome search for endogenous antigens of NKT cells in mammals illustrates a general issue in glycolipid research: the lack of techniques to accurately analyze glycolipids and to synthesize isomerically pure standards[10,15]. Classical (tandem-) mass spectrometry (MS) workflows are sensitive but unable to distinguish the anomeric configuration of GalCer; nuclear magnetic resonance (NMR) yields comprehensive stereochemical information but requires comparably large sample amounts, and cannot ensure the detection of low-abundant isomers in mixtures. Hundreds of different glycosphingolipids were identified in nature based upon sugar heterogeneity, without taking into account the structural diversity of lipid moieties[3]. However, as illustrated by the example of GalCer, a biological function is often related to one specific isomer, and minute alterations of the glycolipid structure can completely eradicate its function. Isomer distinction is thus a highly relevant issue requiring novel analytical approaches.

Here we investigate consistent sets of synthetic glycolipid isomers (Supplementary Table 1) using cryogenic gas-phase infrared (IR) spectroscopy in helium nanodroplets[16]. In this technique, protonated or sodiated glycolipids are generated by nano- electrospray ionization, mass-to-charge selected, pre-cooled by buffer gas cooling (80 K) and then captured in superfluid helium nanodroplets. The latter function as IR-transparent cryostats with an internal temperature of 0.4 K[17]. Upon the resonant absorption of an IR photon by the ion inside a droplet, the vibrational energy is rapidly dissipated by evaporative cooling. After the absorption of multiple photons and repeated cycles of helium evaporation, the bare ion is released from the droplet and detected by MS. IR spectra are generated by monitoring the ion yield while scanning the wavenumber range of interest. The tunable, high intensity IR radiation is provided by the Fritz Haber Institute free-electron laser (FHI FEL)[18]. The technique allows to distinguish not only between α-GalCer and β-GalCer but also between different isomeric glycan headgroups and different lipid moieties. The identification and relative quantification of particular glycolipid isomers in mixtures is demonstrated using synthetic 2-component, 3-component, and 4-component mixtures and two biological lipid extracts from mice.

## Results

### α-galactosylceramide and β-galactosylceramide. The study was initiated by investigating α-GalCer and β-GalCer (Fig. 1a). This

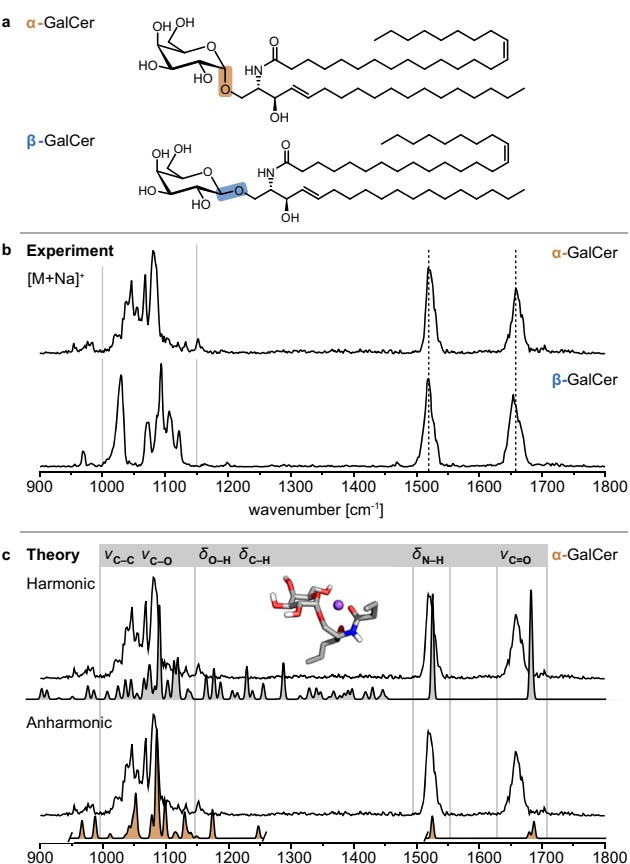

**Fig. 1 Structures and IR spectra of α-GalCer and β-GalCer (d18:1/24:1 (15Z)). a** The stereoisomers α-GalCer and β-GalCer differ by the configuration of the glycosidic bond. **b** Sodium adducts of α-GalCer and β-GalCer yield unique fingerprints in the 1000–1150 cm⁻¹ region. **c** The assignment of vibrational bands is based on computed vibrational spectra of the lowest-energy conformer of [α-GalCer+Na]⁺ with the lipid chains trimmed to four heavy atoms. The anharmonic spectrum (orange) provides a clear match with the experimental fingerprint. Source data are provided as a Source Data file.

pair of stereoisomers is not distinguishable by ion mobility-mass spectrometry (IM-MS) (Supplementary Table 2), and tandem MS relies on relative ion intensities at different collision energies to resolve α-GalCer and β-GalCer[13]. In contrast, cryogenic gas-phase IR spectroscopy probes the ion's structure directly, involving the stereochemistry of the glycosidic bond[16]. The resulting IR spectra therefore feature distinct spectroscopic signatures in the 1000–1150 cm⁻¹ region for α-GalCer and β-GalCer [M+Na]⁺ (Fig. 1b) and [M+H]⁺ ions (Supplementary Fig. 3).

The theoretical spectra of the lowest-energy adducts of [α-GalCer+Na]⁺ and [β-GalCer+Na]⁺ with truncated lipid chains were derived using harmonic approximation (Fig. 1c and Supplementary Fig. 22) and revealed that this diagnostic fingerprint region is composed of C–O and C–C stretching vibrations (ν) of the sugar ring. However, while the absorption frequencies of the non-diagnostic N–H bending vibration (amide II) and the C=O stretching vibration (amide I) are in good agreement with the experimental values, the shape of the spectra in the diagnostic 1000–1150 cm⁻¹ region differs. Reattachment of the lipid chains to a conformer of [α-GalCer+Na]⁺ resulted in only minor changes in the diagnostic region of the theoretical spectrum (Supplementary Fig. 23). Instead, the mismatch originates from the harmonic molecular potential[19] derived using an approximate density

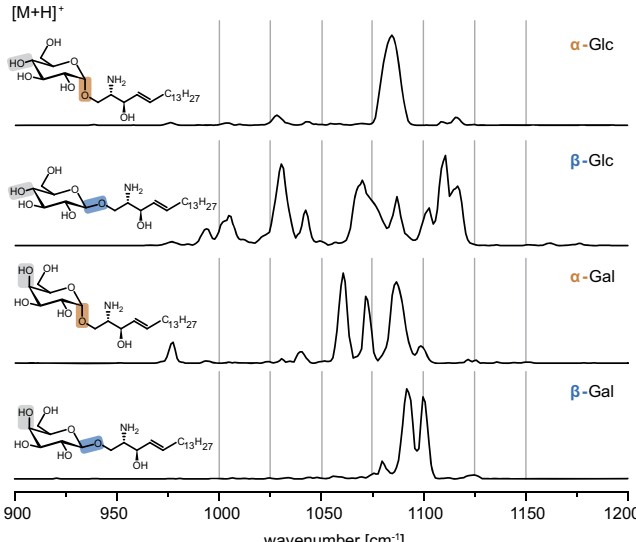

**Fig. 2 Spectroscopic fingerprints of protonated isomeric Gal- and Glc sphingosines.** The highly resolved absorption patterns of each permutation are diagnostic for both the monosaccharide and the configuration of the glycosidic bond. Source data are provided as a Source Data file.

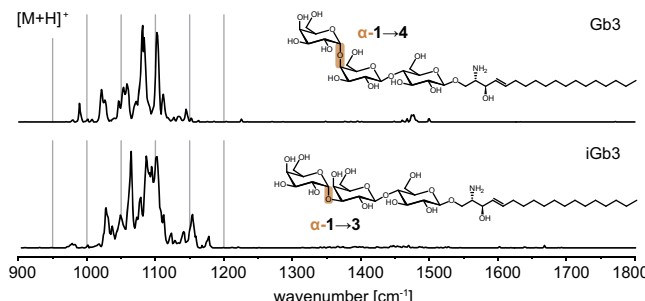

**Fig. 3 IR spectra of isomeric Gb3-sphingosine and iGb3-sphingosine.** The regioisomeric trisaccharides yield distinguishable and well-resolved absorption patterns in the fingerprint region. Source data are provided as a Source Data file.

functional[20], and including anharmonic effects in the theoretical IR spectrum improves the match between the spectra in the fingerprint region (Fig. 1c and Supplementary Fig. 24)[21]. The region between 1150–1450 cm$^{-1}$ is dominated by C–H and O–H bending vibrations ($\delta$) and shows a very low intensity in the experimental spectra. In summary, the spectral signatures of $\alpha$-GalCer and $\beta$-GalCer demonstrate that the assignment of the anomeric configuration can be accomplished exclusively based on the narrow, merely 200 cm$^{-1}$ wide fingerprint region.

**Stereoisomeric monosaccharides.** The ability to distinguish $\alpha$-GalCer and $\beta$-GalCer entailed a more systematic study of isomeric glycolipids, starting with glycosylsphingosines as the simplest possible glycolipids and then gradually increasing size and complexity. The glycosylsphingosines $\alpha$-Gal and $\beta$-Gal sphingosine are the primary degradation products of $\alpha$-GalCer and $\beta$-GalCer formed by the enzymatic removal of the fatty acyl chain[5,22]. It was recently shown that $\alpha$-Gal sphingosine shows antigenic activity towards NKT cells despite the missing lipid chain[22]. In addition to $\alpha$-Gal and $\beta$-Gal sphingosine, the corresponding glucose (Glc) epimers were investigated, as either Glc or Gal are typically linked as first sugars in mammalian glycolipids[1]. Glc and Gal sphingosine are distinguishable after offline-[23] or online[24] modification by tandem-MS but their distinction relies only on relative intensity differences of generated fragments. Without modification, tandem-MS and IM-MS provide no stereochemical information (Supplementary Fig. 2). In contrast, gas-phase IR spectra of the protonated species yield diagnostic, baseline-resolved absorption patterns in the fingerprint region (Fig. 2). The spectra are unique for each combination of monosaccharide (Glc or Gal) and anomeric configuration ($\alpha$ or $\beta$). Some absorption bands are so unique that the corresponding structure could be distinguished from the others by only monitoring the absorption at one specific wavenumber, for example 1065 cm$^{-1}$ for $\alpha$-Gal sphingosine. Besides the diagnostic fingerprint region between 1000–1150 cm$^{-1}$, the spectra display only weak absorption bands associated with the umbrella motion of NH$_3^+$ between 1400–1500 cm$^{-1}$ (Supplementary Fig. 4).

**Regioisomeric trisaccharides.** With increasing glycan size, the number of possible glycan isomers rises exponentially[25] while the spectral resolution deteriorates[16]. To test the influence of the glycan size on the informational content of the IR spectra, globotriose (Gb3) was selected as a common, naturally occurring trisaccharide headgroup containing two Gal and one Glc unit. The ability to resolve subtle structural differences in the trisaccharide was tested by including iso-Gb3 (iGb3), the first reported endogenous NKT cell antigen[26,27]. The chemical structures of Gb3 and iGb3 differ by the connectivity between the two Gal building blocks (1 → 3 vs. 1 → 4) (Fig. 3). This difference in the molecular geometry causes slightly different ion mobilities of sodiated Gb3-sphingosine and iGb3-sphingosine (Supplementary Table 2); however, the individual arrival time distributions of the isomers are not separated in a mixture. Cryogenic IR spectroscopy allows for a much clearer isomer distinction. Even though the IR spectra of Gb3-sphingosine and iGb3-sphingosine are more congested than the spectra of monosaccharide headgroups, they are still well-resolved, and the unique fingerprints demonstrate that the connectivity between monosaccharide building blocks can be determined by IR spectroscopy. Finally, the glycolipid size was further increased by replacing sphingosine by ceramide. The IR spectra of protonated and sodiated $\alpha$-Gb3Cer (d18:1/26:0) display distinct absorption bands but the spectra are more congested, suggesting that the size limit for glycolipids is almost attained (Supplementary Fig. 5).

**Lipid residues.** So far, the investigated glycolipid structures were restricted to sphingolipids based on sphingosine. However, the sphingolipid backbones in nature are not exclusively relying on sphingosine, and a smaller number of glycolipids in mammals are not at all based on a sphingolipid—but a glycerol backbone[1,3]. Several glycerolipids bearing $\alpha$-linked Gal[28] or Glc[29] headgroups were for example identified as bacterial ligands of NKT cells. The influence of different lipid moieties on the IR spectra is shown on the example of $\alpha$-Gal attached to sphingosine, phytosphingosine, ceramide (d18:1/24:1(15Z)) and diacylglycerol (14:0/14:0) (Fig. 4). The formal addition of water to the C=C double bond of mammalian sphingosine to generate its plant analog phytosphingosine does not significantly alter the fingerprint region. The frequencies of the three main bands are identical, whereas a weak absorption at 950 cm$^{-1}$ is only visible in the spectrum of $\alpha$-Gal sphingosine. Ceramide, however, yields a significantly different absorption pattern in the diagnostic fingerprint region and leads to the appearance of characteristic amide vibrations above 1450 cm$^{-1}$. The spectrum of $\alpha$-Gal diacylglycerol displays a less defined fingerprint region and more visible C–H bending vibrations. The ester groups yield additional C=O stretching vibrations above 1700 cm$^{-1}$. Calculation revealed substantial mixing

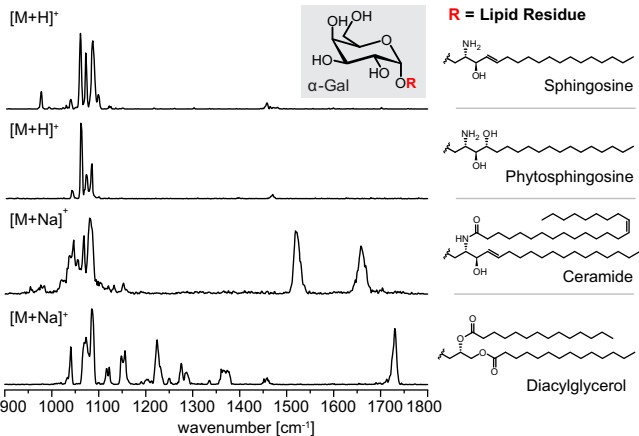

**Fig. 4 Influence of different lipid moieties on the IR spectra of α-Gal lipids.** The exchange of sphingosine for phytosphingosine leads to subtle differences in the spectral fingerprint, whereas ceramide and diacylglycerol considerably alter the fingerprint region. The amide- and ester groups yield additional bands beyond 1450 cm$^{-1}$. Source data are provided as a Source Data file.

of the stretching of the two carbonyl groups resulting in in-phase and out-of-phase vibrational modes (Supplementary Fig. 25). The two modes are, however, not resolved in the experimental spectrum. Overall, these examples highlight the fact that, despite their exceptional resolution, cryogenic IR spectra are challenging to deconvolute using known increments. As a result, routine analyses will require spectral libraries containing distinct glycolipid reference data.

**Glycolipid mixtures.** Reference spectra of glycolipid standards can allow for studying more complex isomeric mixtures and estimating molar ratios. To test the utility of this approach, a proof-of-concept study on several glycolipid mixtures was performed. Three different aspects were addressed: (1) variation of mixing ratios in binary synthetic mixtures to evaluate the dependence of the absorption intensities on the relative concentrations and to determine the limit of detection, (2) deconvolution of more complex synthetic mixtures composed of up to four isomeric glycolipids, and (3) application to biological lipid extracts.

At first, binary mixtures of α-GalCer and β-GalCer with defined mixing ratios were investigated (Supplementary Fig. 9). The experimental spectra were compared with theoretical spectra obtained by weighting and averaging the two reference spectra of pure α-GalCer and β-GalCer according to their mixing ratios. Even though this simple mathematical approach assumes a strictly linear decrease of intensity with decreasing relative concentration, the theoretical spectra model the experimental spectra well (Supplementary Fig. 9d–h). This finding implies that the relative intensities scale roughly linearly with the molar ratio over a wide range of mixing ratios. The respective contributions of the pure compounds to the mixtures were also quantitatively determined with an exceptional accuracy by non-negative matrix factorization (NMF, Supplementary Figs. 10–11)[30,31]. This factorization method deconvolutes the spectra of isomeric mixtures into the spectra of α-GalCer and β-GalCer (basis vectors), and their relative contribution to each of the mixtures (weighting factors). The weighting factors were found to be accurate within an error range of less than 5%. NMF is thus a well-suited method for spectral deconvolution of binary glycolipid mixtures, provided that the abundance of the minor isomer is not much below 5%. In accordance with this limit of reliable detection, α-GalCer could be

detected and quantified in a 5:95 (α:β) mixture but was undetectable in a 1:99 mixture. This detection limit for minor species in isomeric mixtures is within the same order of magnitude as that of NMR spectroscopy[32].

More complex ternary and quaternary mixtures of isomeric glycolipids were investigated to assess if spectral deconvolution is still possible at increasing spectral congestion. For this purpose, the four isomers α-Gal/Glc and β-Gal/Glc phytosphingosine were mixed in any possible combination of 2-component, 3-component, and 4-component mixtures with equal concentrations. NMF correctly retrieves which isomer is present in which of the 11 mixtures (Supplementary Figs. 12–14). Due to the increased complexity of the mixtures (four instead of two possible compounds), the mixing ratios predicted by NMF are not as exact as in the case of binary GalCer mixtures but still sufficiently accurate.

Having established the utility of IR spectroscopy and NMF for the identification and relative quantification of isomers in synthetic glycolipid mixtures, the method was applied to biological lipid extracts. Two lipid extracts **1** and **2** were prepared from cells of α-galactosidase (GLA) and α-glucosidase (GAA) knockout mice, respectively[33,34]. After reversed-phase HPLC separation, glycosylceramides (monoisotopic mass = 809.7 amu) were detected in both samples by MS and MS/MS (Supplementary Fig. 16). Isobaric phosphatidylcholines were removed by treatment with NaOH (Supplementary Figs. 17–18), and the remaining sodiated glycosylceramides were investigated without further purification by IR spectroscopy in the diagnostic fingerprint region and in the amide region (Supplementary Fig. 19). The successful removal of phosphatidylcholines was confirmed by the absence of characteristic ester carbonyl stretching vibrations between 1700–1800 cm$^{-1}$. To assign the isomers present in the biological samples, IR spectra of α-GlcCer and β-GlcCer (d18:1/24:1(15Z)) standards were recorded (Supplementary Fig. 8). Both biological samples display very similar spectroscopic fingerprints with β-GlcCer as predominant isomer (Fig. 5). However, the results from NMF clearly indicate that extract **2**, contrary to extract **1**, also contains a considerable fraction of α-GlcCer (Supplementary Figs. 20, 21). This finding agrees with the fact that extract **2** was obtained from GAA mice, which lack an enzyme cleaving alpha-glucosidic bonds. In Folch extract **1** from GLA mice, the presence of α-GalCer could, however, not be confirmed. In fact, only GlcCer but not GalCer was reliably detected in the biological samples. In general, the accuracy of weighting factors obtained by NMF is higher in the case of synthetic mixtures than for deconvolution of biological mixtures. This can be partly attributed to the lower signal-to-noise ($s/n$) ratio of the spectra caused by a lower glycolipid concentration in the biological extracts. The spectrum of extract **1** displays a higher $s/n$ ratio than the spectrum of extract **2**, which agrees with a higher MS signal intensity (Supplementary Fig. 17 vs. Supplementary Fig. 18).

The exemplary investigation of two biological lipid extracts demonstrates that MS-based IR spectroscopy can provide informative spectra despite low sample concentration and interferences from the biological matrix, while requiring only few basic purification steps. As the number of possible monoglycosyl lipid isomers underlying a certain $m/z$ peak is restricted (usually Glc or Gal), the isomers in question can be unambiguously identified with the help of a small set of reference spectra. The sensitivity of the technique is sufficient to identify certain changes in the isomer distribution, as shown by the example of GAA mice. In contrast to NMR spectroscopy—the gold-standard for direct structure assignment of molecules in solution—IR spectroscopy is furthermore compatible with the small glycolipid quantities typically found in biological samples.

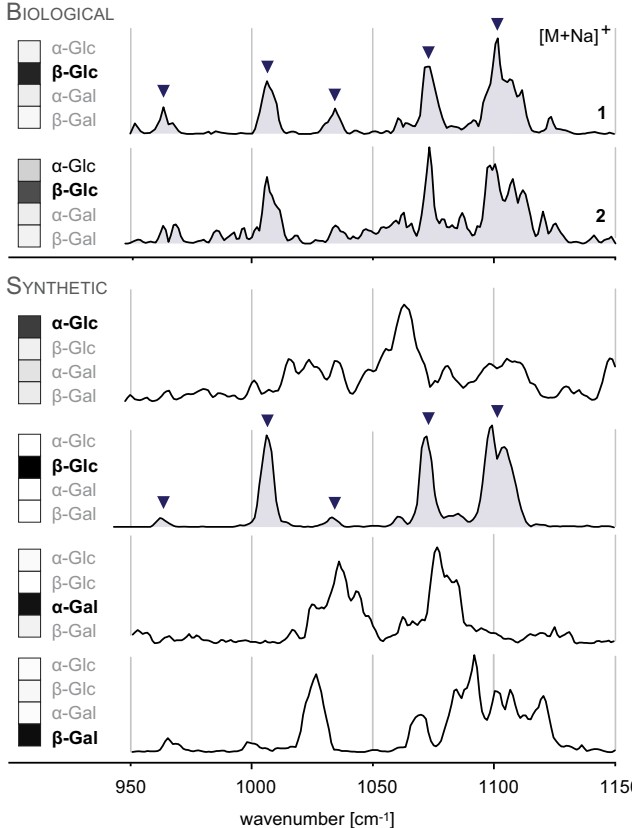

**Fig. 5 IR spectra of sodiated glycosylceramides (*m/z* 832.7) from biological lipid extracts and from synthetic α-Glc/GalCer and β-Glc/GalCer (d18:1/24:1(15Z)) standards.** The five main absorption bands found in both spectra of biological glycolipids are equally present in the spectrum of β-GlcCer. In addition, α-GlcCer contributes to the spectrum of Folch extract **2** isolated from spleen of GAA knockout mice. The isomer contributions obtained from spectral deconvoluted by NMF are depicted in the left panel. Source data are provided as a Source Data file.

Assuming sample concentrations of 0.1–1 mM required for NMR spectroscopy vs. 0.01–0.1 mM for IR spectroscopy, and sample volumes of 1 mL and 10 µL for a complete measurement, respectively, NMR spectroscopy requires a 100–1000 fold larger total sample amount (0.1–1 µmol) than MS-based IR spectroscopy (0.1–1 nmol). Furthermore, and again contrary to NMR, IR spectroscopy does not require pure samples; the quality of IR spectra is not impaired by impurities from biological matrices present in solution, because the glycolipid of interested is isolated in the gas phase by a mass-to-charge filter prior to measurement.

## Discussion

In conclusion, this comprehensive spectroscopic study demonstrates the potential of cryogenic gas-phase IR spectroscopy for the characterization of glycolipid isomers. The technique overcomes substantial analytical difficulties previously expressed with reference to immunological studies, where "the anomeric identity of the isolated compound could not be probed directly by MS given that α-anomers and β-anomers are isobaric species or by NMR because quantities were so limiting"[5]. Using IR spectroscopy, all investigated structural features including anomeric configurations, regioisomeric and stereoisomeric glycan headgroups and different lipid classes were unambiguously resolved. Substantial advantages of the MS-based detection scheme over existing structure-sensitive techniques, such as NMR spectroscopy are indeed a 100–1000-fold lower sample consumption and

tolerance towards non-isobaric impurities, which pave the way for straightforward biological applications without extensive sample purification and enrichment. The practical application is further facilitated by the narrow spectral range covering only 200 cm⁻¹, in which most of the structural information is condensed. Both the high sensitivity and short scan time allowed for the characterization of low-abundant glycolipids from biological lipid extracts and enabled the monitoring of changes in the isomer distribution. The deconvolution of spectra of glycolipid mixtures requires reference spectra from synthetic glycolipids; however, as the number of potential glycolipid isomers in biology is limited, this limitation will resolve over time. Much more complex biological glycan headgroups might become accessible by fragmentation and subsequent spectroscopic interrogation of the generated fragments. Furthermore, gas-phase IR spectroscopy has the potential to become more widely applicable in the future, when tagging spectroscopy techniques that require less powerful, commercially available benchtop light sources are used.

## Methods

**Sample preparation**. β-GlcCer, β-Gb3 sphingosine, and β-iGb3 sphingosine were purchased from Avanti Polar Lipids (Alabaster, USA). Synthesis routes of the investigated glycosyl (phyto-)sphingosines[5] and α-Gb3Cer[35] were described previously. α-GalCer and β-GalCer[36], α-GlcCer[36], α-Gal diacylglycerol[37] and β-Gal diacylglycerol[38] were synthesized by following published procedures and adapting the lipid residues. 100 µM and 10 µM solutions of each glycolipid were prepared for obtaining IR spectra and ion mobility data, respectively. β-Gb3- and β-iGb3 sphingosine were dissolved in pure methanol. α-Gb3Cer was dissolved in dimethyl sulfoxide and diluted with methanol. The other glycolipids were dissolved in dimethyl sulfoxide (1–15 mM) and diluted in a 1:1 (v:v) mixture of acetonitrile and chloroform to obtain 1 mM stock solutions. Prior to measurements, the stock solutions were diluted in a 2:2:1 (v:v:v) mixture of acetonitrile, methanol and water. All solvents were purchased from Sigma-Aldrich. The solutions were stored at −32 °C.

**IM-MS and tandem MS**. Drift tube ion mobility-mass spectrometry (DT-IM-MS) and tandem mass spectrometry (MS/MS) were performed on a modified Synapt G2-S HDMS instrument (Waters Corporation, Manchester, UK) containing a drift tube instead of the commercial traveling wave cell[39]. Glycolipid solutions (10 µM) were prepared as described in the previous section. In addition to protonated and sodiated glycolipids, silver adducts were also investigated. Silver adduction was shown to enable isomer distinction in several lipids by IM[40], and silver ion chromatography is commonly employed to separate lipids due to the preference of Ag⁺ ions to coordinate carbon–carbon double bonds in hydrocarbon chains[41,42]. Silver adducts were prepared by mixing a 17 mM solution of Ag[PF₆] in acetonitrile with 100 µM glycolipid solutions in a ratio of 1:10. Ions were generated by nano-electrospray ionization and drift times were converted into collision cross sections (CCS) using the Mason–Schamp equation[43]. The measurements were repeated on three different days. The double standard deviation of the individual measurements is in all cases ≤1% of the absolute CCS. MS/MS spectra were obtained by collision-induced dissociation (CID) in the trap cell.

**Computational methods**. The conformational space of sodiated glycolipids was sampled using Maestro[44] relying on force field molecular dynamics and CREST[45]. To save computational time and render the conformational search tractable, the lipid chain was truncated to feature the glycan moiety and relevant functional groups of the lipid chain. The sampling of sodiated α-GalCer and β-GalCer was performed using low-mode molecular dynamics sampling in Maestro and Amber* force field and the resulting structures were reoptimized in FHI-aims[46] using PBE + vdWTS [47,48] dispersion-corrected density-functional approximation (DFT) and light basis set settings. This method showed chemical accuracy for a large carbohydrate benchmark set[49–51]. The sampling of sodiated α-Gal and β-Gal diacylglycerol was done using CREST with GFN2-xTB[52] and default settings. A series of low-energy conformers below a threshold of 20 kJ mol⁻¹ were reoptimized for each glycolipid in Gaussian 16 Rev. A.03[53] at PBE0 + D3/6-311 + G(d,p) level of theory[54] and using ultrafine grid settings. Harmonic frequencies were computed at the same level of theory and scaled by a factor of 0.965. For each glycolipid, the lowest-energy structure, which also yields the best spectral match, is shown (Supplementary Figs. 22 and 25). To determine whether truncation of the lipid chains affects the IR signature, full lipid chains were added to one of the low-energy conformers of truncated α-GalCer. The force field-based conformational search was repeated with restraints on atoms of the truncated parent cation. A compact structure with multiple Van der Waals contacts between the lipid chains and the galactose moiety was selected, and reoptimized using the same DFT level of theory and followed by derivation of a harmonic spectrum (Supplementary Fig. 23). In addition, anharmonic spectra of four low-energy conformers of sodiated α-GalCer

were derived using the GVPT2 method implemented in Gaussian 16 Rev. B01[21,55]. These calculations were performed at the same PBE0 + D3/6-311 + G(d,p) level of theory and using ultrafine grid settings for modes 68–88 (1000–1200 cm$^{-1}$ region) and 128–130 (amide II, amide I, and C=C vibrations, respectively), which correspond to the vibrations in the experimental window (Supplementary Fig. 24). The resulting anharmonic spectrum was shifted by a constant factor of 20 cm$^{-1}$.

**Synthetic glycolipid mixtures**. Synthetic isomeric mixtures of GalCer were prepared by mixing 100 μM solutions of α-GalCer and β-GalCer with different mixing ratios: 50:50, 75:25, 90:10, 95:5, and 99:1 (β:α). Mixtures of Glc/Gal phytosphingosines were prepared by mixing 100 μM solutions of the pure isomers to obtain any possible combination of 2-component mixtures (1:1), 3-component mixtures (1:1:1), and a 4-component mixture (1:1:1:1). IR spectra of the protonated glycolipids (GalCer m/z 810.7; Glc/Gal phytosphingosines m/z 480.4) were recorded in the diagnostic fingerprint region (1000–1150 cm$^{-1}$) for the pure isomers and mixtures. The individual measurements were usually performed during one day, and a constant laser focus was applied to reduce variations of the laser fluence. The experimental spectra of GalCer with different mixing ratios were compared to simulated spectra, which were generated by averaging the spectra of the pure isomers with the expected ratios (1:1, 1:3, 1:9, 1:19, and 1:99) using the averaging function in OriginPro 2020 (OriginLab Corporation). Both the resulting simulated spectrum and the experimental spectrum were then normalized to a surface area of 1 in the region from 1000 to 1150 cm$^{-1}$ and superposed (Supplementary Fig. 9d–h).

**Non-negative matrix factorization (NMF)**. NMF factorizes an input matrix into two matrices containing the basis vectors and weighting factors, respectively. In contrast to other factorization methods such as principal component analysis, NMF forces all matrix elements to be non-negative—which is an inherent property of IR data—and therefore only allows for additive combinations of single components[30,56]. In the present work, the input matrix contains experimental IR spectra of glycolipid mixtures, which are deconvoluted into the component spectra and their relative abundance. In the case of binary GalCer mixtures, the input matrix contains five spectra of isomeric mixtures as well as the spectra of the pure isomers. The output is a matrix containing the two spectra of α-GalCer and β-GalCer multiplied by a matrix containing the relative contribution of each isomer to the mixtures. Before applying NMF, the x-axis of the experimental spectra was binned into 76 data points from 1000 to 1150 cm$^{-1}$ (2 cm$^{-1}$ steps) in OriginPro 2020 using the 1D binning application. The obtained input matrices were normalized before applying NMF (Supplementary Table 3). The input matrix of Glc/Gal phytosphingosine mixtures contains 15 spectra comprising the four single component spectra, six 2-component spectra, four 3-component spectra, and one 4-component spectrum. All input spectra were binned into 76 data points from 1000 to 1150 cm$^{-1}$ and normalized following the procedure described above (Supplementary Table 4). The input matrix for deconvolution of biological lipid extracts contains four spectra of sodiated α-Glc/GalCer and β-Glc/GalCer standards and two spectra of sodiated biological glycolipids from extracts **1** and **2**. The spectra were binned into 100 datapoints from 952 to 1150 cm$^{-1}$ (2 cm$^{-1}$ steps) and normalized (Supplementary Table 6). The factorization was carried out using the NMF python module sklearn.decomposition.NMF[57] with the following arguments: init = "random", random_state=0, max_iter=2000, alpha=1, and n_components=2 or 4 (number of isomers contained in the input matrix). The output weighting factors were subsequently converted into percentages. Slight deviations of the predicted weighting factors from the actual mixing ratios, as observed in the case of monoglycosyl phytosphingosines, can be partly ascribed to the shortcoming of NMF that the factorization result is not unique[31]. The single component spectra of each of the four isomers exhibit slightly different absolute intensities. Consequently, a low intensity of a specific absorption band in the mixture spectra can indicate either a low relative abundance of the respective isomer or a higher abundance but low absolute intensity. Because the absolute intensities of the pure spectra are not perfectly modeled by NMF, the relative abundance of some isomers is systematically overestimated (see α-Gal in Supplementary Fig. 13), whereas others are underestimated (see β-Gal in Supplementary Fig. 13); however, within well-acceptable limits.

**Folch's extraction, chromatography and hydrolysis**. Biological lipid extracts were prepared by standard extraction procedures[33,34]: Thymi and spleen from GLA and GAA knockout mice (Jackson Laboratory, California, USA) were harvested from 8 to 12-week-old mice in compliance with ethical regulations and with the approval of the Institutional Animal Care and Use Committee (protocol # 09-0057-4). The samples were frozen at −80 °C until use. Eight samples were pooled for a Folch's extraction that proceeded in four successive steps after homogenization of the tissue using a Polytron: 2:1, 1:1, 1:2 chloroform/methanol, and finally 1:1:1 chloroform/methanol/H$_2$O mixture. The four 20 mL fractions were pooled and lyophilized. The extraction was repeated on the dry pellet before a final lyophilization. The crude lipid extract was purified by reversed-phase HPLC using a Dionex Ultimate 3000 LC system. A Supelco C18 column (2.1 mm × 250 mm, 5 μm) at a constant temperature of 60 °C was used for lipid separation. The mobile phase consisted of 70% isopropanol/22% water/8% methanol, and the system was operated at a flow rate of 0.4

mL min$^{-1}$. The detection was carried out with a UV detector at 205 nm. A total of four measurements with an injection volume of 4 μL were carried out for each sample. Several fractions were collected and examined for the presence of glycolipids by MS and MS/MS. Prior to IR spectroscopy, the dried fractions were redissolved in 100 μL acetonitrile/methanol/H$_2$O (2:2:1), and phosphatidylcholines were removed by adding 1.5 μL of a 1 M aqueous NaOH solution per 50 μL glycolipid solution. IR spectra of m/z 832.7 were recorded when the hydrolysis was completed after 1–2 h (monitored by the disappearance of carbonyl stretching vibrations).

## Data availability
The authors declare that the data supporting the findings of this study are available within the paper and its supplementary information files. Source data are provided with this paper.

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

## Acknowledgements

C.K. is grateful for financial support by the Fonds der Chemischen Industrie and Studienstiftung des deutschen Volkes. K.G. thanks the Fonds National de la Recherche (FNR), Luxembourg, for funding the project GlycoCat (13549747). L.T. and P.B.S. acknowledge funding by the National Institute of Allergy and Infectious Diseases (RO1AI123130). M.M. acknowledges funding by the Army Research Office (W911NF2010271). We furthermore thank Prof. Daniel A. Thomas for advice on NMF-related questions.

## Author contributions

S.D., L.K., and P.B.S. synthesized and prepared glycolipids; P.B.S., G.M., G.v.H., L.T., and K.P. designed and conceived the experiments; C.K., E.M., and K.G. performed the experiments; A.Z. purified biological lipid extracts by HPLC; S.G. and W.S. operated the free-electron laser; K.G. and M.M. performed the theoretical calculations; all authors co-wrote the manuscript.

## Competing interests

The authors declare no competing interests.
