## [Peer Review File · Nature Communications]

Reviewers' Comments:

Reviewer #1:

Remarks to the Author:

In this Communication the Authors claim to achieve characterization of glycolipid isomers by cryogenic gas-phase IR spectroscopy. To this purpose they use a series of isomeric glycan with different head-groups, anomeric configurations and/or lipid moieties and by this approach they get a diagnostic spectroscopic fingerprints in a narrow spectral range.

In the abstract, the author stated the following "The results reveal unprecedented possibilities for the characterization of isomeric glycolipid mixtures". Such a statement, however, is not supported by the results. In fact, my main concern is about the applicability of the proposed method for the analysis of mixtures of isomeric compounds. The authors use standard compounds to prove that validity of the IR method showing that isomeric compounds can be distinguished, however, they did not attempt to analyze mixtures, at different ratios, of such compounds. At this moment this work looks more a style exercise with still no factual potential applications in "real life".

They also did not prove the applicability of the method for the analysis of mixtures of isomeric compounds with unknown structures.

Thus, the applicability of the method is unclear. What is also a little alarming is the fact that despite the potential to distinguish isomers, it is all about 200 cm⁻¹; so a very short zone of the IR spectrum which should be diagnostic of different structural arrangements.

Authors should use samples with mixture of glycolipids and also biological mixtures to prove that this approach can have a real application.

Reviewer #2:

Remarks to the Author:

Reject, but further work might justify a resubmission.

This is a very nice report by Pagel and coworkers on identification of anomers, stereoisomers, regioisomers, and different lipid species of glycolipids by gas-phase IR spectroscopy. They performed cryogenic gas-phase IR spectroscopy in He nanodroplets using a nanoelectrospray ion source, mass spectrometer, and IR free-electron laser. This Reviewer fully agrees with the suggestion of the authors that this IR spectroscopy has a potential ability to analyze isomeric forms of glycolipids very clearly. The experimental IR spectra in this report show different IR features for different isomers, in particular in the 1000-1200 cm⁻¹ region, but these results themselves are not so surprising on the basis of a number of gas-phase IR spectroscopic papers reported from the FELIX facility.

This Reviewer considers that the point of this study would be whether this IR spectroscopy is applicable to real biological samples in solution. As suggested in the Introduction section, other experimental techniques such as MS, IM-MS, and NMR cannot distinguish isomers, or require large amounts of samples, and the authors claimed that this IR spectroscopy could overcome these difficulties. Regarding this point, however, this Reviewer is not completely convinced by this report, because there seem to be many factors that strongly affect the results. This Reviewer has to say that this paper is not suitable for publication in Nature Comm. in the current form, but it would be publishable in the future, if the authors could demonstrate quantitative, practical ability of this IR spectroscopy in the gas phase clearly for analysis of glycolipids in solution. The followings are some points that the authors could consider in the future resubmission. I think that the authors should emphasize the advantage point of this IR spectroscopy over the previous methods much more quantitatively.

(1) In this experiment, the authors used a nanoelectrospray ion source with 100 μM and 10 μM solutions of glycolipids. The amount of the samples in this IR spectroscopy should be compared to that used in NMR spectroscopy.

(2) The authors observed the IR spectra of protonated or sodiated species of glycolipids in the gas phase, but no proof was provided on the efficiency of ion formation for different isomers in the electrospray ion source. Is the relative abundance of isomers for protonated or sodiated species in the gas phase the same as that of non-protonated or non-sodiated forms in solution? Or do the glycolipids exist as protonated or sodiated forms in solution?

(3) The authors should compare the detection limit (1-5%) of this IR spectroscopy with the concentration of low-abundant, endogenous α -GalCer in mammalian cells, for instance. Probably you can also compare the sensitivity of this IR spectroscopy with that of NMR spectroscopy quantitatively.

(4) To identify isomeric forms of glycolipids in practical (biological) samples by this IR spectroscopy, you have to obtain IR spectra of pure (synthesized) glycolipid samples and make database of IR spectra in advance, which seems to be time-consuming processes. At the moment the quantum chemical calculations are not so useful to predict isomeric forms by the comparison between the experimental and calculated IR spectra, as shown in Fig. 1. You need much more theoretical efforts to explain the experimental IR spectra and determine isomeric forms with the IR spectra of practical samples alone, without using the database of IR spectra in the gas phase. I am not so sure if it is no problem to truncate the lipid chains in the theoretical calculations. Long lipid chains will provide many conformers, which show many kinds of IR features.

(5) For practical samples, you probably need to distinguish IR bands of different isomers in one IR spectrum to identify low-abundant isomers. This was already performed by IR-IR double-resonance spectroscopy by Mark Johnson in Yale University or others under cold gas-phase conditions with the messenger technique. The authors mentioned the possibility of using the tagging technique and table-top IR lasers in the future, but it would be better for the authors to emphasize the advantage of using He nanodroplets and the free-electron IR laser, even though your IR technique has difficulties due to non-linear spectroscopic features.

Reviewer #3:

Remarks to the Author:

The aim of this work finds good scientific justification, given the fundamental significance of glycolipids to biochemistry of cell membranes. The known differences between the glycolipid isomers constituting to antigens that appear in less- and more evolutionary advanced animals is critical for our understanding of the functions of immune system in general.

The primary novelty and value of this work stems from application of cryogenic IR spectroscopy, which enables elucidating structural differences between the isomers of immunologically relevant glycolipids.

Having said the above, some of the shortcomings apparent in this work need to be considered.

1. The major weakness of this study results from it being restricted to harmonic approximation in the vibrational analysis. This leads to a less-than-ideal agreement between the theoretical and experimental spectra. Given that the theoretical spectra are the key element of this work, on which the conclusions are based, at least the possible limiting factors need to be critically discussed in this work.

The size of the studied systems is well within the reach of anharmonic methods, as even bigger molecules were already treated successfully with anharmonic methods. Very accurate modelling of anharmonic spectra for even larger molecules was demonstrated in recent literature, and even near-IR region solely populated by overtones and combination bands can be reproduced in very good agreement with experimental spectra. Please refer e.g. to the following paper (doi: 10.3389/fchem.2019.00048).

Further, anharmonic calculations would not only better reproduce the positions of the fundamental bands. Judging from the experimental lineshape in the fingerprint region of glycolipids studied in this work, it seems very likely that the meaningful influence of vibrational resonances can be seen there. Such effects have been shown to be quite significant in IR fingerprint (e.g. refer to the paper: 10.1021/acs.jpca.6b11734).

I understand that within a revision of this manuscript (and further taking into account that this is a

communication paper) it is not feasible to repeat the computational study with anharmonic treatment given the vastly increased demand for time resource. However, it is highly recommended that authors reconsider their discussion taking into account the above-mentioned phenomena.

2. While the scaling of harmonic frequencies will not be able to dismiss the deficiencies of the harmonic approximation in this case (see point 1.), a relatively easily applicable improvement (perhaps forming an additional aid in the discussion) could be evaluated by the authors in the form of a single-parameter scaling (one of the most known method is WLS method), e.g. described here: O. Bak, P. Borowski, Scaling procedures in vibrational spectroscopy In *Molecular Spectroscopy—Experiment and Theory: From Molecules to Functional Materials*, A. Kolezynski, M. Krol (Ed.), Springer, 2018.

Further shortcomings should be corrected in the text:

- l. 130-132; there is no coupling within the harmonic approximation (in other words, the additive character of the harmonic potential – which is the key feature of the harmonic approximation – leads to the coupling potential between any modes being zero). The authors intention is clear in that sentence, however, a different terminology should be used. The concept underlying this sentence is that for the discussed normal mode, there is a substantial mixing of internal coordinates (through which the normal coordinate can be represented) that describe vibrations of the two ester groups. It could also be mentioned, that this observation is still limited by the harmonic approximation, and in fact, it is a harmonic mode under the discussion. However, the latter issue is unlikely to be practically meaningful for this particular study.

As the final remark, it should be mentioned that this study involved a carefully executed conformational search prior the IR spectra modeling, and it should be seen as a strong point of the article.

Having said the above, I believe that this work bears sufficient level of originality and immediate impact, and despite some deficiencies of the methodology, it should be accepted for publication after the Authors re-consider their discussion taking into account the remarks and suggestions given above. The shortcomings in methodology could be addressed in the future disseminations following this communication paper.

Univ.-Prof.Dr. Christian W. Huck

Reviewer #1 (Remarks to the Author):

In this Communication the Authors claim to achieve characterization of glycolipid isomers by cryogenic gas-phase IR spectroscopy. To this purpose they use a series of isomeric glycan with different head-groups, anomeric configurations and/or lipid moieties and by this approach they get a diagnostic spectroscopic fingerprints in a narrow spectral range.

In the abstract, the author stated the following “The results reveal unprecedented possibilities for the characterization of isomeric glycolipid mixtures”. Such a statement, however, is not supported by the results. In fact, my main concern is about the applicability of the proposed method for the analysis of mixtures of isomeric compounds. The authors use standard compounds to prove that validity of the IR method showing that isomeric compounds can be distinguished, however, they did not attempt to analyze mixtures, at different ratios, of such compounds. At this moment this work looks more a style exercise with still no factual potential applications in “real life”. They also did not prove the applicability of the method for the analysis of mixtures of isomeric compounds with unknown structures.

Thus, the applicability of the method is unclear. What is also a little alarming is the fact that despite the potential to distinguish isomers, it is all about 200 cm⁻¹; so a very short zone of the IR spectrum which should be diagnostic of different structural arrangements.

Authors should use samples with mixture of glycolipids and also biological mixtures to prove that this approach can have a real application.

#We thank the reviewer for raising these fully justified concerns about the applicability of the spectroscopic approach for (synthetic and biological) mixtures. The issue of binary synthetic mixtures was briefly addressed in the original manuscript, but we agree with the reviewer that this issue must be extended and further highlighted. The existing data were therefore re-evaluated much more rigorously using a mathematical factorization approach and were complemented by more comprehensive, novel experimental data from both synthetic and biological glycolipid mixtures. The revised manuscript now involves three examples: (1) a synthetic mixture of α/β -galactosylceramides (GalCer) with defined mixing ratios to address the question of relative quantification and sensitivity, (2) synthetic mixtures containing up to 4 isomeric glycolipids to address the issue of multi-component mixtures, and (3) a biological application. Overall, the data show that isomeric glycolipid mixtures can be readily deconvoluted by adapted methods such as non-negative matrix factorization used in this work. Further, biological matrices do not disturb the detection of glycolipids, which makes the method suitable to real-life applications, as shown by the investigation of biological glycolipid extracts in the revised manuscript.

On the other hand, the analysis of completely unknown structures, as proposed by the reviewer, is currently not possible. Theoretical predictions of IR spectra are not yet at the point of yielding correct predictions for the fingerprint region and therefore reference spectra are a crucial requirement for any reliable structural assignment. However, as stated in the revised version of the manuscript, the number of possible isomers of a given mass in biological samples is limited. In mammals, for example, only glucose or galactose can be directly attached to the lipid moiety.^[1] Other structures can therefore be ruled out, leaving only a reduced set of glycolipid standards to be measured for reference purposes. When we measured the biological glycolipid extracts, we had reference spectra of GalCer and α -GlcCer at hand. The spectrum of the biological sample resembled none of those but we knew that β -GlcCer would be the only remaining plausible candidate – and its spectrum matched that of the biological glycolipid, when measured shortly afterwards.

We also agree with the reviewer that the diagnostic spectral region is relatively narrow. However, we see this as a positive point, as it enormously facilitates the data acquisition by reducing the scan time. The reviewer’s concern about spectral congestion is nevertheless justified in the case of glycolipids with more complex glycan structures. For the monoglycosyl glycolipids studied in this work, however, spectral congestion is not an issue. Even the 4-component spectra of glycosylphosphatidylinositols and -ceramides were correctly deconvoluted by non-negative matrix factorization. Owing to buffer gas pre-cooling (i.e. structural annealing) and cryogenic temperatures in the helium droplets, the number of occupied vibrational states is limited and the absorption bands are extraordinarily narrow. Band positions are therefore very exact, and even closely spaced absorption bands can be distinguished. Interferences in the fingerprint region derived from other ions of a slightly different

mass can be ruled out because the experimental setup involves a narrow mass selection window and high-resolution MS detection.

- [1] R. L. Schnaar, T. Kinoshita, in *Essentials of Glycobiology*, 3. ed. (Eds.: A. Varki, R. D. Cummings, J. D. Esko, P. Stanley, G. W. Hart, M. Aebi, A. G. Darvill, T. Kinoshita, N. H. Packer, J. H. Prestegard, R. L. Schnaar, P. H. Seeberger), Cold Spring Harbor Laboratory Press, Cold Spring Harbor (NY), **2015**, pp. 125-135.#

Reviewer #2 (Remarks to the Author):

Reject, but further work might justify a resubmission.

This is a very nice report by Pagel and coworkers on identification of anomers, stereoisomers, regioisomers, and different lipid species of glycolipids by gas-phase IR spectroscopy. They performed cryogenic gas-phase IR spectroscopy in He nanodroplets using a nanoelectrospray ion source, mass spectrometer, and IR free-electron laser. This Reviewer fully agrees with the suggestion of the authors that this IR spectroscopy has a potential ability to analyze isomeric forms of glycolipids very clearly. The experimental IR spectra in this report show different IR features for different isomers, in particular in the 1000-1200 cm⁻¹ region, but these results themselves are not so surprising on the basis of a number of gas-phase IR spectroscopic papers reported from the FELIX facility.

This Reviewer considers that the point of this study would be whether this IR spectroscopy is applicable to real biological samples in solution. As suggested in the Introduction section, other experimental techniques such as MS, IM-MS, and NMR cannot distinguish isomers, or require large amounts of samples, and the authors claimed that this IR spectroscopy could overcome these difficulties. Regarding this point, however, this Reviewer is not completely convinced by this report, because there seem to be many factors that strongly affect the results. This Reviewer has to say that this paper is not suitable for publication in Nature Comm. in the current form, but it would be publishable in the future, if the authors could demonstrate quantitative, practical ability of this IR spectroscopy in the gas phase clearly for analysis of glycolipids in solution. The followings are some points that the authors could consider in the future resubmission. I think that the authors should emphasize the advantage point of this IR spectroscopy over the previous methods much more quantitatively.

#We thank the reviewer for the overall positive evaluation of our manuscript. We agree that the applicability of the technique to biological glycolipids must be proven and compared to other analytical techniques. As stated in the comments to reviewer 1, we have met the main demand of both reviewers to demonstrate the utility of IR spectroscopy for real-life biological glycolipid samples after a preceding proof-of-concept study on the relative quantification of isomers in synthetic mixtures. In addition, we further elaborated the advantages of IR spectroscopy over other structure-sensitive analytical techniques, in particular NMR spectroscopy, by following the advices below.#

(1) In this experiment, the authors used a nanoelectrospray ion source with 100 μM and 10 μM solutions of glycolipids. The amount of the samples in this IR spectroscopy should be compared to that used in NMR spectroscopy.

#We thank the reviewer for this very helpful advice to highlight the utility of our technique for the analysis of small glycolipid quantities as typically found in biological matrices. These sample quantities are indeed usually insufficient for NMR analyses. We therefore included a short paragraph at the end of the manuscript, comparing the total sample amounts required for NMR and for IR spectroscopy by a simple calculation example. While the sample concentrations used for IR spectroscopy and NMR are still comparable, the total sample amounts required for one measurement are vastly different. For the acquisition of full-scan IR spectrum, we need less than 10 μL sample solution, while NMR requires roughly 1 mL. This calculation suggests 100-1000 times less total sample required for IR spectroscopy, which is also highlighted in the final discussion with respect to biological applications. In addition, we included a brief discussion on the generally high tolerance of gas-phase IR spectroscopy towards sample impurities. #

(2) The authors observed the IR spectra of protonated or sodiated species of glycolipids in the gas phase, but no proof was provided on the efficiency of ion formation for different isomers in the electrospray ion source. Is the relative abundance of isomers for protonated or sodiated species in the gas phase the same as that of non-

protonated or non-sodiated forms in solution? Or do the glycolipids exist as protonated or sodiated forms in solution?

#We thank the reviewer for raising this point. Glycolipid isomers generally exhibit a comparable ionization efficiency because they carry the same functional groups (hydroxyl groups, amides or amines), which determine the proton affinity and ability to coordinate metal cations such as Na⁺. This assumption is supported by the fact that we observe very similar absolute ion intensities for the single isomers (at the same concentration), as well as comparable relative intensities of protonated and sodiated species (see Figure 1, obtained with the same nESI source). The relative amount of protonated and sodiated species mostly depends on the ions present in solution, rather than the stereochemistry of the glycolipid isomers; a high content of sodium salts, for example, suppresses the formation of protonated species. As the isomers contained in an isomeric mixture have the same concentration of Na⁺ cations available, the relative amount of isomers in the form of [M+Na]⁺ ions should reflect the distribution of the isomers in solution. In solution, the glycolipids do not occur as either sodiated or protonated forms but are dynamically surrounded by solvated cations and anions. The protonated and sodiated species are formed during the electrospray process.^[2] An exception are glycosyl sphingosines, where a considerable fraction is probably constantly protonated at the primary amine in solution, depending on the pH. In conclusion, we are convinced that differences in the ionization efficiency do not occur or are negligible for the stereoisomeric glycolipids studied here. If ionization efficiencies differed between the isomers, the spectral deconvolution of synthetic mixture spectra would not reproduce the expected and well-known mixing ratios.

Figure 1. MS spectra of alpha- and beta-GlcCer obtained under identical conditions on a Synapt G2-S HDMS instrument. The absolute ion intensities as well as relative intensities of [M+H]⁺ ions (*m/z* 810) and [M+Na]⁺ ions (*m/z* 832) are comparable between the isomers.

[2] L. Konermann, E. Ahadi, A. D. Rodriguez, S. Vahidi, *Anal. Chem.* **2013**, *85*, 2-9.#

(3) The authors should compare the detection limit (1-5%) of this IR spectroscopy with the concentration of low-abundant, endogenous α -GalCer in mammalian cells, for instance. Probably you can also compare the sensitivity of this IR spectroscopy with that of NMR spectroscopy quantitatively.

#We thank the reviewer for this question. Unfortunately, the concentration of endogenous α -GalCer is not known to date, as there are so far no reliable methods for quantification. Future studies using IR spectroscopy might fill this gap. However, we were able to detect a small amount of α -GlcCer in the lipid extract from GAA-

knockout mice (Figure 5), showing that potential biological applications exist despite the detection limit. The sensitivity of IR spectroscopy in terms of detecting minor isomers in isomeric mixtures is comparable to that of NMR, where a limit of 3–5 %^[3] was found for oligosaccharides. We added a corresponding reference in the revised manuscript when determining the limit of detection for α -GalCer in β -GalCer.

[3] J. Hofmann, H. S. Hahm, P. H. Seeberger, K. Pagel, *Nature* **2015**, *526*, 241.#

(4) To identify isomeric forms of glycolipids in practical (biological) samples by this IR spectroscopy, you have to obtain IR spectra of pure (synthesized) glycolipid samples and make database of IR spectra in advance, which seems to be time-consuming processes. At the moment the quantum chemical calculations are not so useful to predict isomeric forms by the comparison between the experimental and calculated IR spectra, as shown in Fig. 1. You need much more theoretical efforts to explain the experimental IR spectra and determine isomeric forms with the IR spectra of practical samples alone, without using the database of IR spectra in the gas phase. I am not so sure if it is no problem to truncate the lipid chains in the theoretical calculations. Long lipid chains will provide many conformers, which show many kinds of IR features.

#The reviewer raises a valid point whether quantum calculations are able to offer a viable alternative to the time-consuming process of obtaining biological samples. Long lipid chains will increase both the system size and conformational space to be sampled substantially, which will result in an exponential increase of computational time needed to generate a reliable spectrum. In order to understand how the truncation of the lipid chains and neglect of the anharmonic effects affect the simulated spectra, we investigated in detail a low-energy conformer of $[\alpha\text{-GalCer+Na}]^+$.

An optimized cation with truncated lipid chains was first appended with full lipid chains. Then, we performed a force field-based conformational search with restraints on atoms of the truncated parent cation. This search resulted in over 100 different conformers that differed only in the orientation of the lipid chains. From these structures, we selected a compact conformer which presented a lot of Van der Waals contacts between the lipid chains and the galactose moiety to emphasize the potential impact of the lipid chain on the vibration of the diagnostic region. The cation was reoptimized using the same DFT level of theory as for the truncated model and a harmonic spectrum of the ion was computed from the optimized structure.

The resulting spectrum of $[\alpha\text{-GalCer+Na}]^+$ with full lipid chains is compared with its truncated version in Figure S23. The simulations show that the impact of the lipid chains on the shape of the spectrum is marginal. The most intense C-O bands in the diagnostic region of 1000–1150 cm^{-1} are blue-shifted on average by 5 cm^{-1} due to interactions with the lipid chains (which results in narrowing of the harmonic potential) but the overall shape of the spectrum remains unaffected meaning that the lipid chains do not contribute to the diagnostic IR features. Furthermore, because the main driving force for the adduct formation in the gas phase are electrostatic interactions between the sodium cation and the oxygen atoms, full lipid chains will have limited impact on the coordination site. This argues that truncation of the lipid chains is a reasonable simplification during derivation of the theoretical IR spectrum.

In order to improve the match in the diagnostic region between the theoretical and experimental spectrum, we included anharmonic effects. Figure S23 confirms that anharmonic effects change the spectrum in the fingerprint region. Evaluation of the anharmonic spectra of four low-energy conformers of the truncated $[\alpha\text{-GalCer+Na}]^+$ adduct (Figure S24) indeed improved the match between the lowest-energy conformer and the experimental spectrum in the diagnostic region between 1000–1150 cm^{-1} . This example shows that such calculations, although computationally demanding, might in the future offer a feasible alternative for the biological sampling.#

(5) For practical samples, you probably need to distinguish IR bands of different isomers in one IR spectrum to identify low-abundant isomers. This was already performed by IR-IR double-resonance spectroscopy by Mark Johnson in Yale University or others under cold gas-phase conditions with the messenger technique. The authors mentioned the possibility of using the tagging technique and table-top IR lasers in the future, but it would be better for the authors to emphasize the advantage of using He nanodroplets and the free-electron IR laser, even though your IR technique has difficulties due to non-linear spectroscopic features.

#We agree with the reviewer that we should not point out the non-linearity of our technique and therefore deleted the corresponding sentence in the discussion. In fact, the mathematical analysis of synthetic mixture

spectra by non-negative matrix factorization showed that relative absorption intensities of the different isomers in the mixture scale rather linearly with the relative abundance. Contrary to our expectations, the helium droplet technique seems to be very well applicable for relative quantification of glycolipids in isomeric mixtures with the help of reference spectra, which is demonstrated and highlighted at several instances in the revised manuscript. However, we regard as important to refer to tagging spectroscopy, which could indeed make gas-phase IR spectroscopy more commonly applicable and is therefore a central selling point of our study. Experiments with the FEL are suitable for proof-of-principle studies but not attractive for research groups who do not have regular access to such a light source. Therefore, we mention tagging spectroscopy combined with tabletop lasers at the end of the discussion, however, this time not in the context of linearity of absorption but in the context of future applicability.#

Reviewer #3 (Remarks to the Author):

Reviewer #3 (Remarks to the Author):

The aim of this work finds good scientific justification, given the fundamental significance of glycolipids to biochemistry of cell membranes. The known differences between the glycolipid isomers constituting to antigens that appear in less- and more evolutionary advanced animals is critical for our understanding of the functions of immune system in general.

The primary novelty and value of this work stems from application of cryogenic IR spectroscopy, which enables elucidating structural differences between the isomers of immunologically relevant glycolipids.

Having said the above, some of the shortcomings apparent in this work need to be considered.

#We thank the reviewer for highlighting the significance of the present manuscript and address further issues regarding the vibrational analysis in more detail below.#

1. The major weakness of this study results from it being restricted to harmonic approximation in the vibrational analysis. This leads to a less-than-ideal agreement between the theoretical and experimental spectra. Given that the theoretical spectra are the key element of this work, on which the conclusions are based, at least the possible limiting factors need to be critically discussed in this work. The size of the studied systems is well within the reach of anharmonic methods, as even bigger molecules were already treated successfully with anharmonic methods. Very accurate modelling of anharmonic spectra for even larger molecules was demonstrated in recent literature, and even near-IR region solely populated by overtones and combination bands can be reproduced in very good agreement with experimental spectra. Please refer e.g. to the following paper (doi: 10.3389/fchem.2019.00048).

Further, anharmonic calculations would not only better reproduce the positions of the fundamental bands. Judging from the experimental lineshape in the fingerprint region of glycolipids studied in this work, it seems very likely that the meaningful influence of vibrational resonances can be seen there. Such effects have been shown to be quite significant in IR fingerprint (e.g. refer to the paper: 10.1021/acs.jpca.6b11734).

I understand that within a revision of this manuscript (and further taking into account that this is a communication paper) it is not feasible to repeat the computational study with anharmonic treatment given the vastly increased demand for time resource. However, it is highly recommended that authors reconsider their discussion taking into account the above-mentioned phenomena.

#We thank the reviewer for bringing our attention to recent developments in the anharmonic calculations of theoretical IR spectra.

The comparison between experimental and theoretical spectra is indeed a notoriously challenging task that suffers from (1) incomplete sampling of the conformational space, limited basis sets, density-functional approximations and harmonic approximation on the theory side; (2) discrepancy between computed electric dipoles and how the absorption in the action spectroscopy is measured; and (3) lack of clear measure of similarity of two spectra. Although disentangling these effects on the IR-fingerprint region of biologically relevant molecules reaches beyond the scope of this communication, we decided to reinvestigate the match between the

theoretical and experimental spectrum of $[\alpha\text{-GalCer+Na}]^+$, which showed poorer agreement with the experiment than the beta anomer.

We followed the reviewer's suggestion and evaluated anharmonic spectra of four low-energy conformers using GVPT2/PBE0+D3/6-311+G(d,p) level of theory. For these calculations we selected 21 bands that occur in the diagnostic region between 1000–1200 cm^{-1} and amide I, amide II, and C=C vibrations (24 bands in total for each molecule). All derived spectra are shown in Figure S24. The anharmonic corrections clearly improve the match between the spectrum of the lowest-energy conformer and the experimental spectrum, especially in the diagnostic region between 1000–1200 cm^{-1} . The anharmonic spectrum, added to Figure 1c, demonstrates that we can confirm the structure of the ion using the diagnostic region. In addition to the revised figure, the relevant paragraph in the manuscript has been adjusted and the Supporting Information has been updated. #

2. While the scaling of harmonic frequencies will not be able to dismiss the deficiencies of the harmonic approximation in this case (see point 1.), a relatively easily applicable improvement (perhaps forming an additional aid in the discussion) could be evaluated by the authors in the form of a single-parameter scaling (one of the most known method is WLS method), e.g. described here: O. Bak, P. Borowski, Scaling procedures in vibrational spectroscopy In *Molecular Spectroscopy—Experiment and Theory: From Molecules to Functional Materials*, A. Kolezynski, M. Krol (Ed.), Springer, 2018.

#We thank the reviewer for the suggestion. It is clear that the application of custom scaling parameters improves the overall match between the experimental and theoretical spectra. The suggested WLS method presents a viable ansatz for this problem, however as discussed in the point 1, we decided to investigate the anharmonic fingerprints of $[\alpha\text{-GalCer+Na}]^+$ instead.

Regarding the other ions for which harmonic spectra are presented in the SI, we prefer to keep the same uniform scaling parameter of 0.965 which is consistent with our previous work: Mucha et al. *Nature Communications* 2020, 9, 4174 and Marianski et al., *Angew. Chem. Int. Ed.* 2020, 59, 6166—even at the price of poorer agreement in the 1000–1200 cm^{-1} region. We observe that this parameter provides an excellent match between theoretical and experimental spectra for the amide bands. Furthermore, such scaling factor is consistent with other scaling factors of IR spectra of biomolecules simulated with hybrid functionals that have been reported throughout the literature. #

Further shortcomings should be corrected in the text:

- l. 130-132; there is no coupling within the harmonic approximation (in other words, the additive character of the harmonic potential – which is the key feature of the harmonic approximation – leads to the coupling potential between any modes being zero). The authors intention is clear in that sentence, however, a different terminology should be used. The concept underlying this sentence is that for the discussed normal mode, there is a substantial mixing of internal coordinates (through which the normal coordinate can be represented) that describe vibrations of the two ester groups. It could also be mentioned, that this observation is still limited by the harmonic approximation, and in fact, it is a harmonic mode under the discussion. However, the latter issue is unlikely to be practically meaningful for this particular study.

#We thank reviewer for pointing out our misleading phrasing. Because the sentence is primarily intended to provide a description of the vibrations, the respective part in the manuscript has been adjusted from:

“Quantum-chemical calculations revealed that the vibrations of the two ester groups are coupled; they either stretch in- or out-of-phase while being both coordinated to the sodium cation”

to:

“Calculation revealed substantial mixing of the stretching of the two carbonyl groups resulting in in-phase and out-of-phase vibrational modes.” #

As the final remark, it should be mentioned that this study involved a carefully executed conformational search prior the IR spectra modeling, and it should be seen as a strong point of the article.

Having said the above, I believe that this work bears sufficient level of originality and immediate impact, and despite some deficiencies of the methodology, it should be accepted for publication after the Authors re-

consider their discussion taking into account the remarks and suggestions given above. The shortcomings in methodology could be addressed in the future disseminations following this communication paper.

Once again, we would like to thank the reviewer for the helpful comments regarding anharmonic calculations that improved the overall quality of the manuscript.#

Univ.-Prof.Dr. Christian W. Huck

Reviewers' Comments:

Reviewer #1:

Remarks to the Author:

I am fully satisfied with authors rebuttal, their further experiments and their discussion post revision. therefore I think that no this is a very fine piece of work and that this deserves publication.

Reviewer #3:

Remarks to the Author:

Checking the manuscript step-by-step compared to the original submission it gives the impression that authors have put huge energy and a big effort to fullfill all demands from the reviewers. The doubts araised by all three reviewers concerning the applicability, suitability and reliability for real biological samples could be shown somehow.

Authors explain the superior performance to other more conventional analytical approaches, including mass spectrometry, NMR. These methods have the advantage of being extremely well established, fully supported by high technological performance and data interpretation support. It is well-known in the field of analytical chemistry that gas-phase IR spectroscopy is one of the techniques, that might have a high chance to become one of the future emerging technologies. Now I could discuss all individual and detailed points araised by reviewer 1,2 and 3. This will bring for all of them the same answer. Authors answered were possible with all possibilities being available at the current state of the art. Of course such new applications and developments can never be 100% perfect (which is also not really necessary), it gives provides a huge step forward with this new type of analytical approach. Balancing the doubts of reviewers with the answers from the authors I have the impression, that this manuscript will be worth being published in Nature Communications. Therefore, I recommend to accept it.